

# Analysis of new particle nucleation events and comparisons to simulations of particle number concentrations based on GEOS-Chem/APM in Beijing, China

Kun Wang[1], Xiaoyan Ma[1*], Rong Tian[1], Fangqun Yu[2]

[1]Key Laboratory for Aerosol-Cloud-Precipitation of China Meteorological Administration, Nanjing University of Information Science & Technology, Nanjing 210044, China

[2]Atmospheric Sciences Research Center, University at Albany, Albany, NY, USA.

*Correspondence to*: Xiaoyan Ma (xma@nuist.edu.cn)

**Abstract.** Aerosol particles play important roles in air quality and global climate change. In this study, we analyzed the measurements of particle size distribution from March 12th to April 6th, 2016 in Beijing to characterize new particle formation (NPF) by using the observational data of sulfuric acid, meteorological parameters, solar radiation, and $PM_{2.5}$ mass concentration. During this 26-day campaign, 11 new particle formation events were identified with obvious bursts of sub-3 nm particle number concentrations and subsequent growth of these nucleated particles. It is found that sulfuric acid concentration in Beijing did not have a significant difference between NPF and non-event days. Although the temperature during NPF days in Beijing was slightly higher than that on non-event days, temperature was not necessarily the key factor to determine NPF because higher solar radiation intensity usually increases the temperature. Low relative humidity (RH) and high daily total solar radiation appeared to be favorable to the occurrence of NPF events, which was more obvious in this campaign. A quantitative analysis indicated that more than 90% of NPF events occur when the daily total solar radiation was greater than 19 MJ/m²/day and RH was less than 26.5%. The $PM_{2.5}$ mass concentration can also be used as a rough and simple criterion to predict the occurrence of NPF events. In addition, the simulations using four nucleation schemes, i.e., $H_2SO_4$-$H_2O$ binary homogeneous nucleation (BHN), $H_2SO_4$-$H_2O$-$NH_3$ ternary homogeneous nucleation (THN), $H_2SO_4$-$H_2O$-ion binary ion-mediated nucleation (BIMN), and $H_2SO_4$-$H_2O$-$NH_3$-ion ternary ion-mediated nucleation (TIMN), based on a global chemistry transport model (GEOS-Chem) coupled with an advanced particle microphysics (APM) model, were conducted to study the particle number concentrations and new particle formation process. Our comparisons between measurements and simulations

indicate that BHN scheme and BIMN scheme significantly underestimated the observed particle number concentrations, and

the THN scheme captured well the total particle number concentration on most NPF event days but failed to capture the

noticeable increase in particle number concentrations on March 18th and April 1st. TIMN scheme had obvious improvement

in terms of total and sub-3 nm particle number concentrations and nucleation rates. This study provides a basis for further

understanding of new particle nucleation mechanism in Beijing.

**1 Introduction**

New particle nucleation is the process by which gaseous pollutants transform into particles (gas-particle reaction), which

is the main source of particle number abundance generated by secondary transformation in the atmosphere (Yu et al., 2008;

Merikanto et al., 2009). The process that the newly generated fine particles grow into particles with larger particle size through

further collision, condensation and moisture absorption is well known as new particle formation (NPF) event.

With the development of industry and the increase of human activities, air quality is deteriorating day by day, and

atmospheric particulate matter has become one of the major sources of air pollutants. Particulate matter not only affects the

radiation balance by scattering and absorbing solar radiation, but also indirectly modifies the climate by acting as cloud

condensation nuclei (CCN) (Merikanto et al., 2009). In addition, new particles can also impact the formation of fog and haze.

Some studies found that in the context of the decline in pollutant emissions caused by the lockdown during the COVID-19

epidemic, new particles derived from NPF played a significant role in the formation of haze (Huang et al., 2020; Tang et al.,

2021; Li et al., 2021). Guo et al. (2014) found that after an NPF event, smog-haze pollution continued to occur in Beijing,

indicating that aerosol nucleation and growth led to the development of $PM_{2.5}$. When the concentration of atmospheric

particulate matter is high, atmospheric visibility will be reduced in fog and haze weather, and high concentration of atmospheric

fine particulate matter will also harm human health (Kaiser, 2005) as tiny aerosol particles can enter the human body through

the respiratory system.

At present, the nucleation and growth of new particles has been widely studied around the world. Sipilä et al. (2010)

demonstrated a positive relationship between nucleation rate and the sulfuric acid concentration. Kulmala et al. (2013)

established a framework of atmospheric nucleation at three different scales below 2 nm based on observations, identifying the

participation of sulfuric acid in the nucleation process and emphasizing the important role of organic compounds in

atmospheric aerosols. A positive correlation between nucleation rate and sulfuric acid concentration (or $H_2SO_4$ proxy) was also

found in many NPF studies in China (Xiao et al., 2015; Dai et al., 2017).

Besides sulfuric acid, a number of other nucleating precursors, including atmospheric ions, amines, ammonia, iodine

oxides, and organic acids, have been proposed to be involved in the formation of the critical nucleus under different ambient

environments. Amines and ammonia are crucial in NPF because of their ability to stabilize sulfuric acid clusters by forming

acid-base complexes (Yao et al., 2018). Shen et al. (2021) found the growth rate of ions was larger than that of neutral particles

during the COVID-19 lockdown period in Beijing. Currently several major theories have been proposed to explain the

phenomenon of new particle nucleation in the atmosphere, including the classical binary nucleation theory (Hussein, 2005)

such as $H_2SO_4$-$H_2O$ binary nucleation (Kulmala et al., 1998), $H_2SO_4$-$NH_3$-$H_2O$ ternary nucleation (Korhonen et al., 1999), ion-

mediated nucleation (Yu and Turco, 2000), organic compounds participated nucleation (Wang et al., 2015) and so on. Recently,

Wu et al. (2020) proposed a possible physical mechanism to explain NPF in China. They found that though there are differences

in chemical emissions over various regions, there is a common relationship between the characteristics of NPF and stability

intensity, that is unstable atmospheric turbulence will effectively reduce condensation sink (CS) by diluting pre-existing aerosol

particles and so as to promote nucleation.

There are some special features associated with NPF events found in China. For example, the concentrations of nucleating

precursors and pre-existing aerosol particles both can be quite high in polluted cities, which is different compared with cleaner

environments (Kulmala et al., 2016). Jayaratne et al. (2017) conducted the observations in Beijing during the winter of 2015

and did not observe any NPF event when the daily mean $PM_{2.5}$ concentrations were higher than 43 $\mu g/m^3$. However, in some

cases, the condensation sinks or the average coagulation sinks were not significantly lower during NPF events compared to

non-NPF times, suggesting that other factors, such as the precursor vapor and photochemical activity, may also play an

important role in driving NPF (Gong et al., 2010). Beijing is a representative city in northern China because of its developed

industry and commerce, and thus quite high concentrations of atmospheric precursors and pre-existing aerosol particles. Wu

et al. (2007) found that NPF days accounted for 40% of the observation days from March 2004 to February 2005 in Beijing,

which are usually sunny and dry days. Continuous and comprehensive long-term observations would helpful to understand the

mechanism of NPF, and answer the key participants and processes of NPF under complex air conditions in China. Besides, laboratory experiments and model simulations would also be very necessary. Chen et al. (2019) investigated the effect of organics involved nucleation scheme on NPF in Beijing, but previous model studies and comparisons based on different nucleation schemes in Beijing are still lacking. Thus, it is necessary to further examine the issues on nucleation mechanism. The paper is organized as below. We first analyze the favorable background of new particle formation in Beijing, and then obtain the quantitative meteorological and solar radiation conditions of new particle formation. In addition, we conducted the simulations using four nucleation schemes, i.e., $H_2SO_4$-$H_2O$ binary homogeneous nucleation (BHN) (Yu, 2007, 2008), $H_2SO_4$-$H_2O$-$NH_3$ ternary homogeneous nucleation (THN) (Yu, 2006a), $H_2SO_4$-$H_2O$-ion binary ion-mediated nucleation (BIMN) (Yu, 2006b), and $H_2SO_4$-$H_2O$-$NH_3$-ion ternary ion-mediated nucleation (TIMN) (Yu et al., 2018), based on a global chemistry transport model (GEOS-Chem), and compared with the observed NPF events from the measurements from March 12th to April 6th, 2016 in Beijing to understand the nucleation mechanism.

## 2 Methodology

### 2.1 Observations

The observational data used in this paper include particle size distributions, temperature (T), relative humidity (RH), and sulfuric acid concentration, which was provided by Cai et al. (2017a). The measurement field was located on the top floor of a four-story building in the center of the campus of Tsinghua University in Beijing, and all observations were collected during the period from March 12th, 2016 to April 6th, 2016. A home-made diethylene glycol scanning mobility particle spectrometers (DEG-SMPS) was used to measure 1-5 nm particle size distributions and the particle size distributions from 3 nm to 10 μm (in aerodynamic diameter, Liu et al., 2016) were measured by a particle size distribution system (including a TSI aerodynamic particle sizer and two parallel SMPSs, equipped with a TSI nano-DMA and a TSI long DMA, respectively) with a resolution of 5 minutes. A specially designed miniature cylindrical differential mobility analyzer (mini-cyDMA) for effective classification of 1-3 nm (sub-3 nm) aerosol was equipped with the DEG-SMPS (Cai et al., 2017b). Sulfuric acid was measured by a modified high-resolution time-of-flight chemical ionization mass spectrometer (HR-ToF-CIMS, Aerodyne Research Inc.)



with a resolution of 5 minutes. Further details of the measurement methods can be found in Cai et al. (2017a).

In addition, the daily PM$_{2.5}$ mass concentration provided by the China National Environmental Monitoring Center (http://www.cnemc.cn/), and solar radiation datasets, including daily maximum solar radiation flux density and the daily total

solar radiation, measured from the National Meteorological Information Center of the China Meteorological Administration (http://data.cma.cn), are also employed in this study.

**2.2 Model description**

The GEOS-Chem model is a global three-dimensional chemical transport model driven by assimilated meteorological observations from the Goddard Earth Observing System (GEOS) of the NASA Global Modeling Assimilation Office (GMAO).

The GEOS-Chem model includes a detailed simulation of tropospheric O$_3$-NO$_x$-hydrocarbon chemistry as well as aerosols and their precursors (Park et al., 2004). An advanced particle microphysics (APM) model has been coupled with GEOS-Chem to study detailed particle formation and growth processes in the global atmosphere (Yu and Luo, 2009). The APM model is an advanced multi-type, multi-component, size-resolved microphysics code developed for explaining atmospheric particle observations (e.g., Yu, 2006b; Yu and Luo, 2009; Yu and Turco, 2011). The basic microphysical processes in APM include

nucleation, coagulation, condensation/evaporation, thermodynamic equilibrium with local humidity, and dry deposition. In GEOS-Chem/APM, sulfate (or secondary) particles are represented by 40 sectional bins with dry diameters ranging from 1.2 nm to 12 μm, including 30 bins for 1.2–120 nm range and another 10 bins for 0.12–12 μm. In the model, new particle formation is parameterized by ion-mediated nucleation (IMN) which is based on physics and constrained by laboratory data (Yu, 2006b) and can finely predict global nucleation distributions with a reasonable consistency (Yu et al., 2008). In addition, the IMN

takes into account the complex interactions among small air ions, neutral and charged clusters of various sizes, precursor vapor molecules, and pre-existing aerosols. The new particle formation and particle number concentrations predicted by the GEOS-Chem/APM model have been extensively evaluated against a wide range of aircraft-, land-, and ship- based field data (Yu and Luo, 2009, 2010; Yu et al., 2010). In recent years, the APM model has also been coupled with other numerical models such as Nested Air Quality Prediction Modeling System (NAQPMS) and Weather Research and Forecast/Chemistry (WRF-Chem)

model to study new particle formation (Luo and Yu, 2011; Chen et al., 2019).

In this study, we use a version of GEOS-Chem (v12.6.0) driven by the GEOS-FP-assimilated meteorological field, with a spatial resolution of 2° × 2.5° and 47 vertical levels during the period of March 1st, 2016 to April 7th, 2016. We focus on four nucleation schemes to compare simulations with particle number concentration measurements from March 12th to April 7th in this study.

**3 Results and discussions**

**3.1 Occurrence of new particle formation event**

NPF generally includes the following two steps: (1) condensable vapors are produced via chemical reactions then gaseous vapors form critical clusters through gas-particle reaction, (2) the critical clusters continue to grow to a larger size (1-2 nm) (Kulmala et al., 2013) and compete with pre-existing particles to survive in the collision and removal processes at the same time, in this dynamic equilibrium process, new particles can only be formed when growth is dominant.

Currently, there is no unique mathematical criterion or definition for NPF events. Dal Maso et al. (2005) proposed a criterion to justify NPF events, i.e., a new mode of particles start in the nucleation-mode size range, and prevail within a few hours and show signs of growth. The days without the particles in the nucleation-mode are called non-event days. It is noted that some days are not easily classified as either NPF days or non-event days, so these days are usually classified undefined days.

The criteria to determine NPF in this study are as follows: (1) high concentrations of sub-3 nm clusters/particles appear over a time of hours at the onset of the event, (2) subsequent growth to larger sizes for a few hours. In this case, the nucleated particles continued to grow, causing a "banana shape" of an NPF to appear in the particle number size distribution.

Figure 1 shows the particle size distributions observed during March 12th, 2016-April 6th, 2016 in Beijing. According to the criteria of NPF events defined above, 11 typical event days and 13 non-event days are identified during the 26-day campaign. The other 2 days, i.e., March 19th and 30th, the sub-3 nm particle number concentration was relatively low and the evolution of particle size distributions was not continuous, were identified as undefined days. The NPF events were observed with a high frequency (42%) in spring in this campaign, which is similar to the previous study in Beijing (Wu et al., 2007) in which they





found that spring is usually the season with the highest frequency of NPF events in northern China. One explanation is that

stronger wind from the north removes pre-existing particles to create a clean regime, which further leads to the occurrence of

NPF events (Wu et al., 2008; Cai et al., 2017a; Chu et al., 2019; Wu et al., 2020).

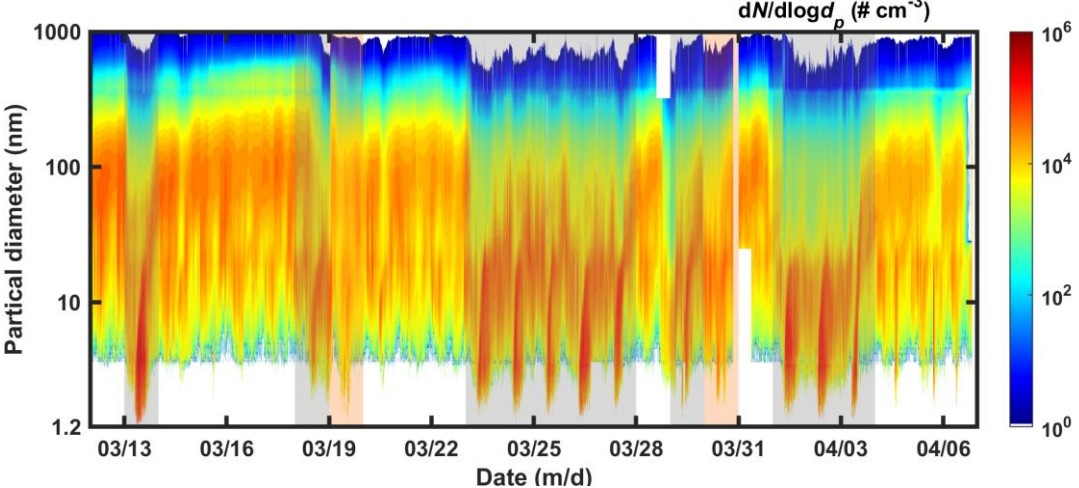

**Figure 1. Contour of measured particle size distributions from March 12 to April 6, 2016. The identified 11 NPF days and 2 undefined days are shadowed by grey and orange background, respectively.**

**3.2 Analysis based on observations**

**3.2.1 The sulfuric acid concentration on NPF days**

Since the number concentration of new particulate matter in the atmosphere is strongly dependent on the concentration

of sulfuric acid, so sulfuric acid is a key precursor species for atmospheric nucleation and has an important contribution to

NPF events (Sipilä et al., 2010). However, it was not easy to measure the concentration of sulfuric acid a few years ago. With

the development of technology, chemical ionization mass spectrometry (CIMS) was used to detect gaseous sulfuric acid in the

atmosphere (Zheng et al, 2011, 2015).

The gaseous sulfuric acid in the atmosphere is mainly produced by the oxidation reaction of sulfur dioxide and OH.

Stockwell and Calvert (1983) demonstrated the reaction as follows:

$$SO_2 + OH \xrightarrow{M} HOSO_2 \tag{1}$$

$$HOSO_2 + O_2 \rightarrow HO_2 + SO_3 \tag{2}$$





$$SO_3 + H_2O \rightarrow H_2SO_4 \tag{3}$$

Figure 2 shows the average diurnal concentration of sulfuric acid on the NPF event days and non-event days. It is known

that NPF events usually occur within a few hours after sunrise and end in the afternoon. During the start period, the averaged

sulfuric acid concentration on NPF days is close to that on non-event days, and compared to NPF days, the averaged sulfuric

acid concentration on non-event days was not significantly low during the whole time period. As shown in Table 1, the average

sulfuric acid concentration during non-event periods is obviously higher than that between 6:00 and 18:00 on NPF days.

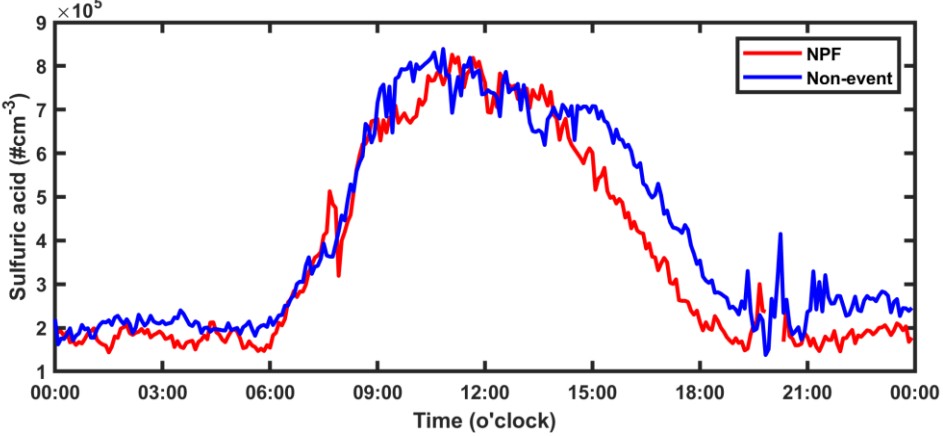

**Figure 2. Averaged diurnal cycles of sulfuric acid concentrations for NPF days and non-event days during the 26-day measurement period.**

**Table 1. Mean sulfuric acid concentrations in the daytime of NPF and non-event days, respectively.**

| Classification | Averaged sulfuric acid concentration (#/cm$^3$) |
| --- | --- |
| NPF days | 567530 |
| non-event days | 591270 |



As illustrated in Figure 3, the gaseous $H_2SO_4$ concentrations exhibit significant diurnal variation, and sulfuric acid concentration closely follows the diurnal solar cycle because of its short atmospheric lifetime (less than 1 min) (Zheng et al., 2011). Compared to another campaign in summer in Beijing (Wu et al., 2020), the daily maximum sulfuric acid concentration measured in this study is relatively low. This might be caused by the relatively weak solar radiation intensity in springtime measurement compared with summertime observation. In Figure 3, the differences among the daily maximum sulfuric acid concentrations were small. For example, the daily maximum sulfuric acid concentrations on some NPF days were not significantly higher than those on non-event days as expected. From April 4th to April 6th, the sulfuric acid concentrations were high but there was no NPF event. Therefore, it is shown that the sulfuric acid concentration in Beijing is sufficient to lead to NPF, which is consistent with some earlier studies in Nanjing (An et al., 2015; Qi et al., 2015) and Shanghai (Xiao et al., 2015). There is usually enough $SO_2$ for NPF to occur under heavily polluted conditions (Herrmann et al., 2014). The observed $SO_2$ concentrations during October 2015 showed that the most polluted area was located in Northern China (Hu et al., 2022). The surfaces of atmospheric aerosols correspond to the major sink for gas-phase sulfuric acid, but an increase in atmospheric sulfuric acid does not always result in more frequent NPF events (Zhang et al., 2012). So far, the presence of gaseous sulfuric acid in concentrations exceeding $10^5$ molecules $cm^{-3}$ has been shown as a necessary condition to observe NPF in the atmosphere (Weber et al., 1999; Nieminen et al., 2009). Overall, the previous results seem to indicate that adequate sulfuric acid concentration was not a limiting factor for NPF in Chinese megacities even if it was necessary.

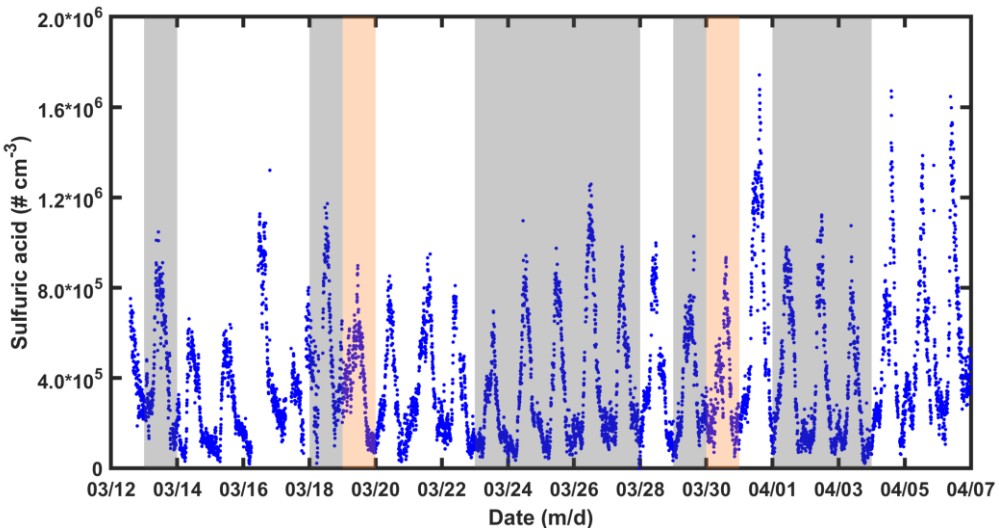

**Figure 3. Time series for the sulfuric acid concentration. The identified NPF days and undefined days are shadowed by grey and orange background, respectively.**

### 3.2.2 Solar radiation and meteorological conditions for NPF

Air pollutants and meteorological conditions are usually studied together with nanoparticles and their precursors. By comparing the pollution characteristics between NPF event and other non-event days, we can obtain some clues on key factors affecting NPF events. As shown in Figure 4, there were few NPF days when the daily total solar radiation was low, but there were 9 NPF days when the daily total solar radiation in Beijing was above 20 MJ/m$^2$/day during the whole observation period. It implies that variation of solar radiation could influence new particle formation via photochemical process (Shen et al., 2021). Strong solar radiation favors the OH formation by photochemical reaction, then OH involves in processes of sulfate formation and VOCs oxidation, and produces sulfuric acid and vapors with low volatility in the atmosphere. Finally, these vapors with low volatility and sulfuric acid molecules can condense together to form new clusters and lead to NPF (Zhang et al., 2012). Previous studies showed that sulfuric acid concentration in Beijing was sufficient to lead to NPF (Wang et al., 2011; Cai et al., 2017a), and the averaged sulfuric acid concentrations on non-event days were not significantly lower than those on NPF days in our study. Laboratory studies and field observations have indicated that organic vapors participate in the particle nucleation besides sulfuric acid (Metzger et al., 2010; Yao et al., 2018). In addition, low volatility organic compounds were proposed to

contribute to the particle growth rate (Tröstl et al., 2016). Sulfuric acid was enough to explain the observed growth for particles

205   smaller than 3 nm but was insufficient to explain the observed growth rates of large particles (Xiao et al., 2015; Yao et al.,

2018). Therefore, when solar radiation was high, the low volatility organic vapors generated by VOCs oxidation may promote

the particle growth. Besides, it is proposed that solar radiation may create a turbulence flow and then strengthen the source

during nucleation process and reduce the sink in the growth process (Wu et al., 2020).

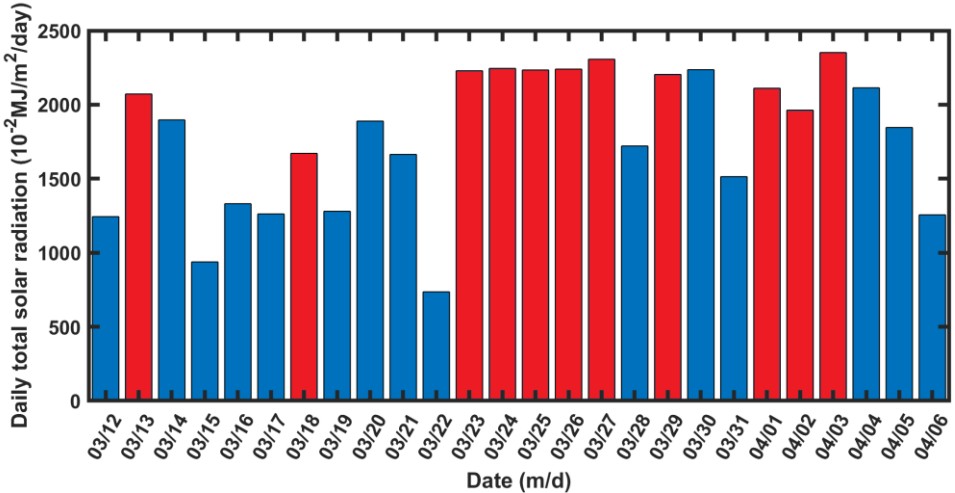

210   **Figure 4. Daily total solar radiation from March 12 to April 6, 2016. The red and blue bars represent the NPF days and other days, respectively.**

Higher solar radiation intensity usually increases the temperature, so the average temperature on NPF days (14.5°C) from

8 to 16 o'clock was 7% higher than that on non-event days (13.6°C) during the same time period (Figure 5). The high

temperature on NPF days indicates that NPF events in Beijing were generally accompanied by high temperature, but the role

of temperature on NPF was inconsistent in many studies. For example, Qi et al. (2015) found that NPF was accompanied by a

higher temperature in spring, summer and autumn of 2011-2013 in Nanjing, but Sun et al. (2021) found that temperature had

no significant role in NPF in the summertime and wintertime in the coastal area of Qingdao, China. It seems that the

temperature associated with NPF events in various cities/regions have different characteristics. In this study, it is indicated that

temperature was not necessarily the key factor to determine NPF events, even if the average temperature during NPF days is

relatively high.





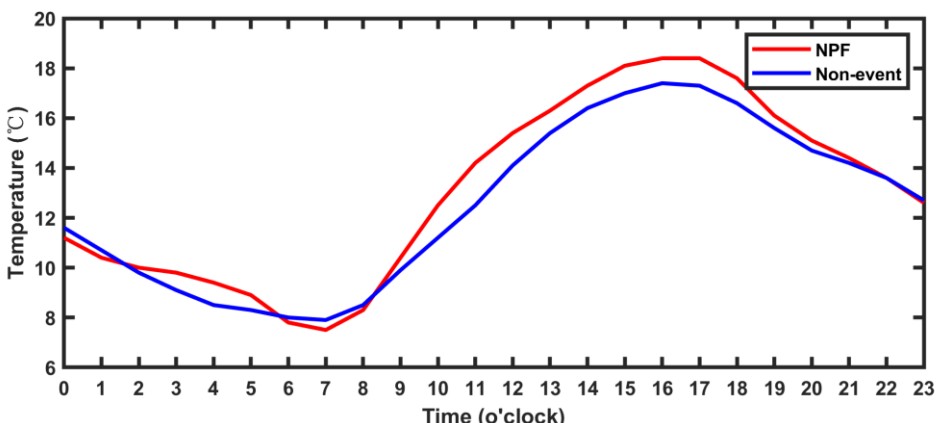

**Figure 5. Averaged diurnal cycles of temperature for NPF days and non-event days during the 26-day measurement period.**

Figure 6 shows the scatter plots of the daily total solar radiation and relative humidity (RH) color coded with $PM_{2.5}$ mass

concentration. The daily values were averaged between 8:00 and 16:00 because NPF events usually occur during the daytime.

The statistics indicate that the average RH for the 11 NPF days and 13 non-event days was 16% and 37%, respectively, implying

that NPF events in Beijing likely occur on days with low relative humidity. The daily total solar radiation of NPF in Beijing

was above 19 MJ/m$^2$/day for 10 out of 11 NPF days, so NPF events generally do not occur on a cloudy day even when pre-

existing particle concentration is low (Hamed et al., 2007). The explanation is possibly because that at high RH, coagulation

scavenging of sub-3 nm clusters was enhanced, while reduced solar radiation led to gas phase oxidation chemistry was

diminished, and also condensation sink of condensable gases was increased due to hygroscopic growth of the pre-existing

particles (Hamed et al., 2011). Besides, a negative correlation between $PM_{2.5}$ mass concentration and the occurrence of NPF

events shown in Figure 6 also indicates that NPF mainly occurs when the $PM_{2.5}$ concentration and gas pollutant concentrations

were both low (Wu et al., 2007). High $PM_{2.5}$ concentrations suppress NPF by increasing the sinks of vapor responsible for

nucleation and growth of clusters and nucleation mode particles (Zhou et al., 2020). Previous study found that new particle

formation is typically completely suppressed when the aerosol surface area exceeds 100 $\mu m^2/cm^3$ (Aalto et al., 2001). Cai et

al. (2017a) found that the Fuchs surface area (which is a representative parameter of coagulation scavenging based on kinetic

theory and is proportional to CS) fundamentally determined the occurrence of NPF events in Beijing. There is a good

correlation between the Fuchs surface area and the $PM_{2.5}$ mass concentration. On NPF days, the $PM_{2.5}$ concentrations between



8:00 and 16:00 were typically lower than 30 µg/m³, except for the event on March 18th, so the $PM_{2.5}$ concentrations can be

used as a rough and simple criterion to predict the occurrence of NPF events. However, an empirical $PM_{2.5}$ threshold value of

30 µg/m³ might not be valid during the whole year, because $PM_{2.5}$ mass concentrations vary significantly with seasons.

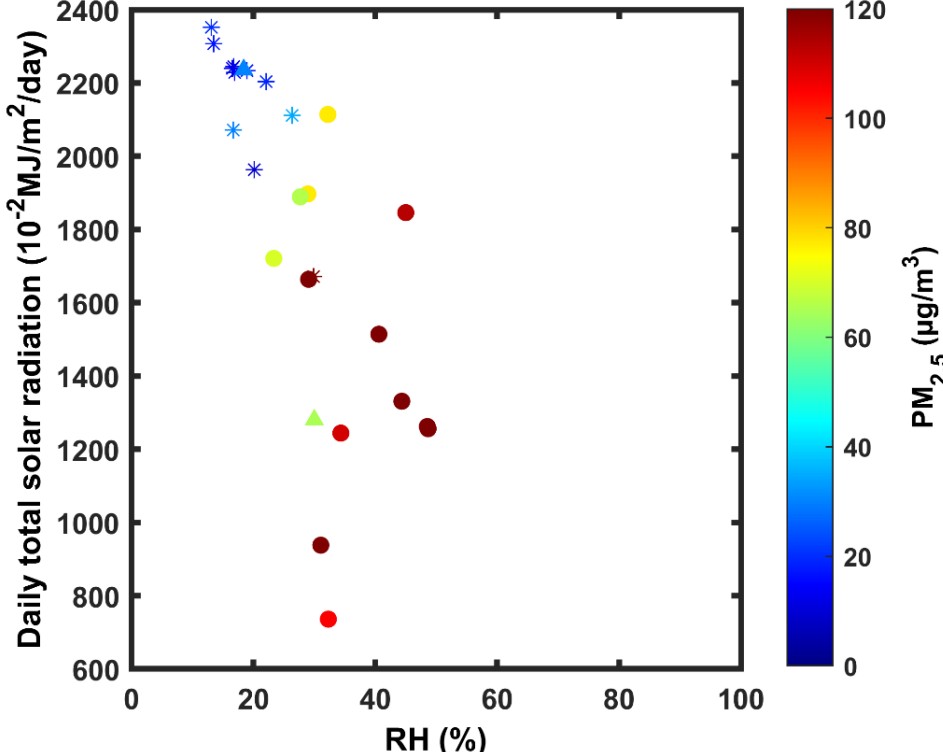

**Figure 6. Scatter plot of NPF days (asterisks), non-event days (circles), and undefined days (triangles) between daily total radiation and RH color coded with PM2.5 mass concentration.**

It is found that the favorable conditions on NPF days were similar, which were significantly different from the conditions

on non-event days. Therefore, it is possible to determine whether there is an NPF event based on a quantitative analysis of

relative humidity and daily total solar radiation. By setting a restricted condition, a high proportion of NPF events were

screened out under the condition (Table 2). When the daily total solar radiation was greater than 19 MJ/m²/day and RH was

less than 26.5%, there were 11 days, including 10 out of 11 NPF days, and NPF events accounted for 91%. Whether the

quantitative empirical conclusion is valid in the long term, it is necessary to check with more records of NPF events in Beijing

in other seasons.





**Table 2. Proportion of NPF days under restricted conditions.**

|  | Daily total radiation above 19 MJ/m$^2$/day and RH below 26.5% |
| --- | --- |
| Number of NPF days | 10 |
| Number of eligible days | 11 |
| Percentage | 91% |





### 3.3 GEOS-Chem/APM model simulation

In order to understand the particle nucleation mechanism in polluted areas such as Beijing, and reduce uncertainties in

model simulations and predictions, we conducted the simulations from four nucleation schemes based on GEOS-Chem/APM.

To ensure the input meteorological parameters are reasonably good and thus reduce the impact on the simulated nucleation,

we firstly compared the first layer meteorological fields of GEOS-FP (about 70 m) input to GEOS-Chem/APM with the

observed meteorological data (about 10 m), including temperature and relative humidity (Figure 7). It is shown that the

temporal variations of the simulated temperature and humidity were quite consistent with that of the observations, though the

magnitude of the simulated temperature was slightly lower than the observed ones.

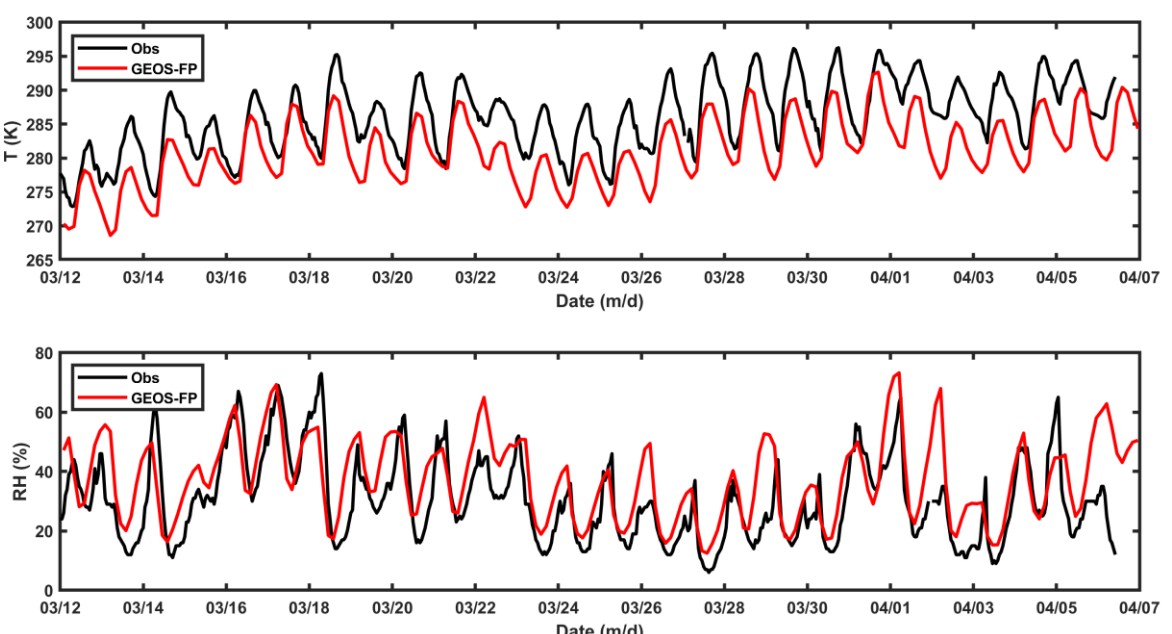

**Figure 7.    Time series of temperature and RH input to model (the first layer, about 70 meters) and time series of measurements at a height of about 10 meters during the 26-day campaign.**

Figures 8 and 9 present the total and sub-3 nm particle number concentrations from the GEOS-Chem/APM simulations

of four nucleation schemes, as well as the observed particle number concentrations, respectively. We found that the simulations

with BHN and BIMN schemes significantly underestimated the observed total particle number concentrations, and thus did




not show significant number concentration fluctuation to distinguish NPF event and non-event event days. The simulation with

the THN scheme showed the total number concentration fluctuation on most NPF event days but failed to capture the noticeably

increase of particle number concentrations on March 18th and April 1st. The simulation with the TIMN scheme reproduced

quite well the increase of particle number concentration on all NPF event days, including continuous NPF events on March

23rd to 27th and every discontinuous single NPF event day, but underestimated the observed daily maximum particle number

concentration on March 26th, 27th and April 1st.

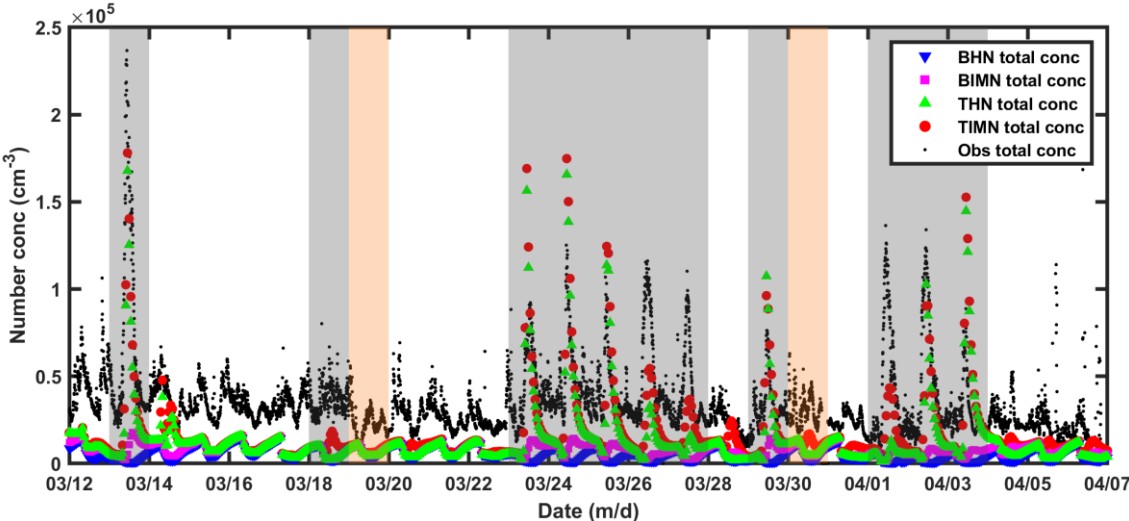

**Figure 8. Comparison of observed total particle number concentrations with those simulated on the basis of BHN, BIMN, THN, and TIMN schemes during the measurement period. The identified NPF days and undefined days are shadowed by grey and orange background, respectively.**





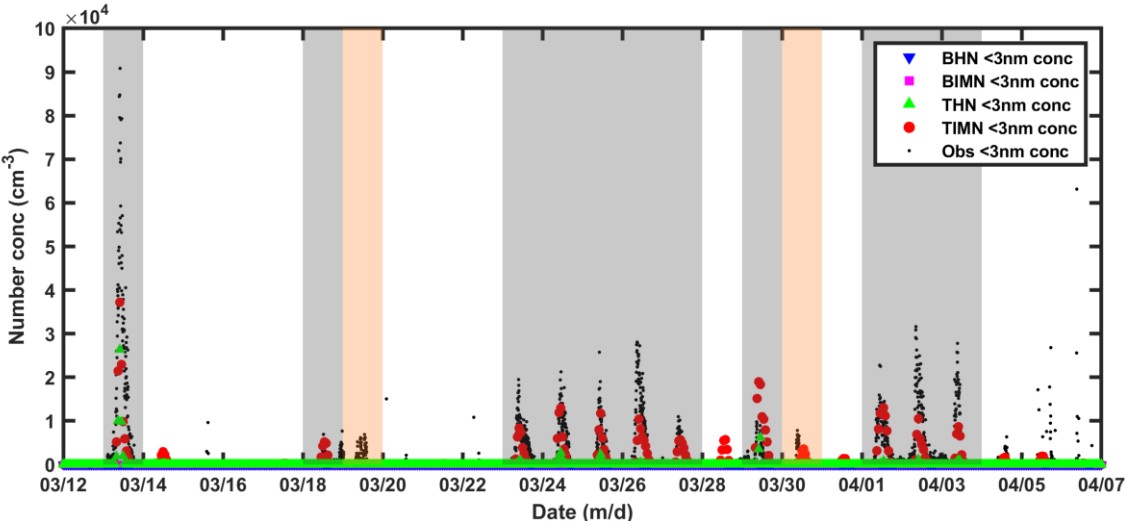

**Figure 9. Comparison of observed sub-3 nm particle number concentrations with those simulated on the basis of BHN, BIMN, THN, and TIMN schemes during the measurement period. The identified NPF days and undefined days are shadowed by grey and orange background, respectively.**

The sub-3 nm particle number concentrations (Figure 9) and nucleation rates (Figure 10) from the simulations with BHN and BIMN were quite close to zero and not sensitive to daily variations. Such low nucleation rates lead to low particle number concentrations in the consequent growth process. The simulated nucleation rates based on THN scheme were obviously lower than those from TIMN scheme, but the total particle number concentrations were not as low as those based on BHN and BIMN scheme. Vertical variations of nucleation rate on April 3rd (Figure 11) indicate that although the THN nucleation rates on the ground were quite small, the nucleation rates in the upper boundary layer were much close to the results with the TIMN scheme. Thus, it is suggested that the downward mixing of particles in the upper boundary layer may contribute to the increase of particle number concentrations on the surface. Yu et al. (2020) found that nucleation rates with TIMN agreed with the well-controlled Cosmics Leaving Outdoor Droplets (CLOUD) measurements within the uncertainties under nearly all conditions. Besides, the comparisons of horizontal spatial distributions of annual mean nucleation rates in the lower boundary layer (0-0.4km) simulated with six different nucleation schemes indicated that annual mean nucleation rate based on the BHN scheme significantly underestimated particle number concentrations (Yu et al., 2010). In contrast, the nucleation rates based on the TIMN scheme were high not only on NPF days, but also on some non-event days. Not all these days with high nucleation rates have high particle number concentrations as well. Even so, the eventual simulated particle number concentrations show good





agreement with observations, which probably indicates that the TIMN scheme has a good simulation performance on the

growth, condensation, coagulation and other processes after the nucleation process.

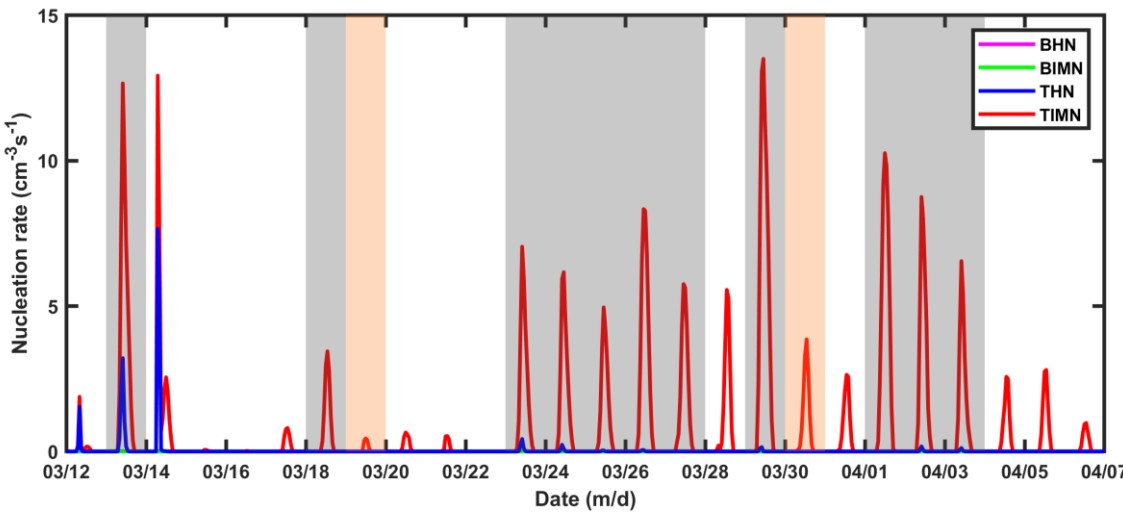

**Figure 10. Time series of nucleation rates simulated on the basis of BHN, BIMN, THN, and TIMN scheme during the measurement period. The identified NPF days and undefined days are shadowed by grey and orange background, respectively.**





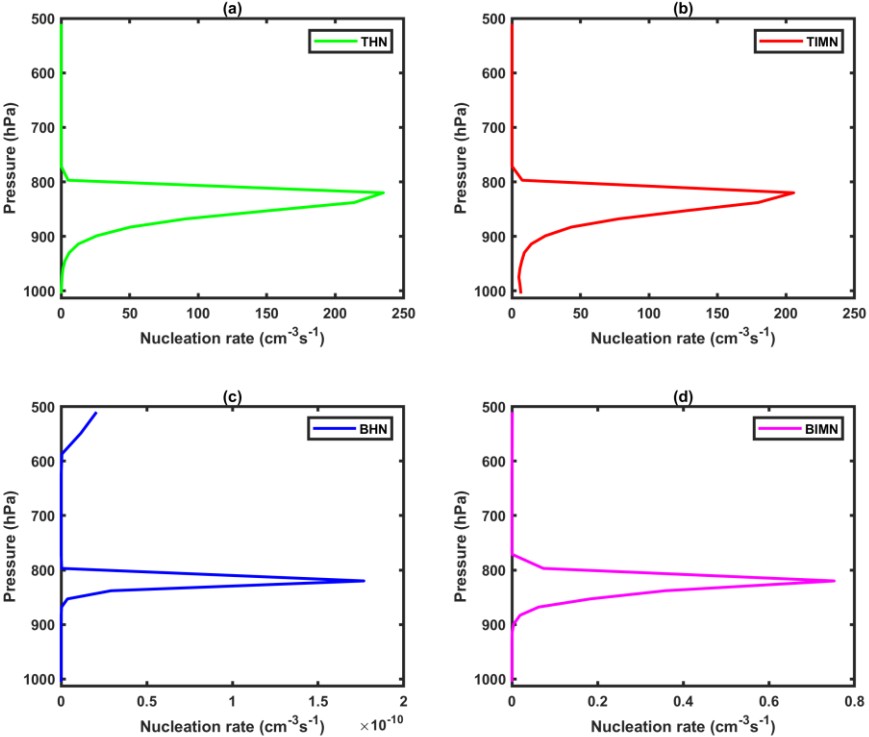

**Figure 11. Vertical variations of nucleation rate below 500hPa simulated with (a) THN, (b) TIMN, (c) BHN, and (d) BIMN scheme at 10:00 am on April 3rd in Beijing.**

Table 3 shows the correlation coefficients between the simulated particle number concentrations and the observed ones

during the entire study period and NPF event days. The simulation with BHN and the observation was negatively correlated, indicating that there were obvious differences between the simulated results of BHN and the observed total particle number concentrations. According to the correlation coefficients, the BIMN scheme also performed not well. In contrast, the particle number concentrations with THN and TIMN were positively correlated with the observations, and the correlation coefficients between observations and simulations with TIMN scheme were higher than those with THN, indicating that TIMN has a better

performance to simulate NPF events. The correlation coefficients between sub-3 nm particle number concentrations and simulation results based on TIMN scheme (about 0.79) during two periods were significantly higher than those between the total particle number concentrations and simulations (about 0.65).



**Table 3. Correlation coefficient of observation total and sub-3nm particle number concentrations with BHN, BIMN, THN and TIMN scheme, respectively.**

|  | Obs_3nm | | Obs_total | |
| --- | --- | --- | --- | --- |
|  | R[a] | R[b] | R[a] | R[b] |
| BHN | -0.016 | 0.006 | -0.202[**] | -0.357[**] |
| BIMN | 0.134[**] | 0.099 | 0.016 | -0.006 |
| THN | 0.773[**] | 0.782[**] | 0.594[**] | 0.586[**] |
| TIMN | 0.795[**] | 0.793[**] | 0.649[**] | 0.657[**] |

[a] During the measurement period; [b] during NPF days; [**] the correlation coefficient passes the statistical significant test ($p < 0.01$).

Figure 12 shows the temporal evolution of simulated particle number size distribution based on the TIMN scheme during 26 days in Beijing. According to the same criterion, all NPF days can be identified, which is consistent with observations. On most NPF days, the simulated number concentrations and particle number size distributions are close to observations. However, the simulated particle number size distributions on non-event days are obviously different from Figure 1, because the

background number concentration is difficult to simulate. Thus, the simulated number concentrations on most non-event days in Figure 8 are indeed significantly lower than observations. One NPF event on March 28th predicted by the model was not observed with measurements, and the simulated particle number concentrations on some NPF days (e.g., March 27th and April 1st) were obviously underestimated. It may suggest that the model would not be able to capture effects at a given site during certain periods when the measurements were affected by sub-grid-scale processes (emissions, plumes, etc.). It may be helpful

to compare high-resolution simulations with observations in order to address the issue.



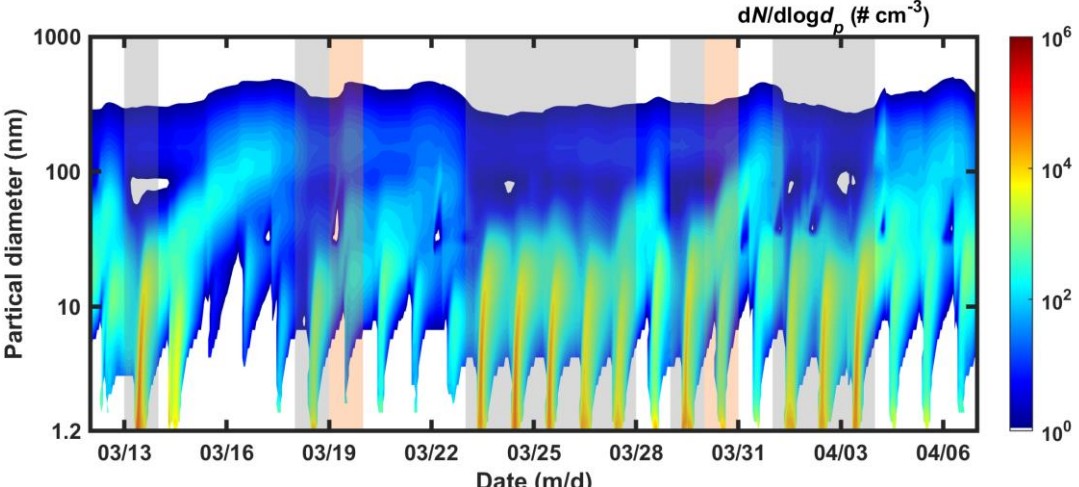

**Figure 12. The simulated particle size distributions based on the TIMN scheme. The identified NPF days and undefined days are shadowed by grey and orange background, respectively.**

The time series of the simulated PM$_{2.5}$ and PM$_{10}$ mass concentrations are compared against the measurements during the study period (Figure 13). It is shown that APM generally reproduces the temporal variations of PM$_{2.5}$ and PM$_{10}$ mass concentrations, but tends to underestimate high values of PM$_{2.5}$ and PM$_{10}$ mass concentrations, possibly due to relatively low spatial resolution. Nevertheless, the correlation coefficients between simulations and observation PM$_{2.5}$ and PM$_{10}$ mass concentrations reached 0.619 and 0.496, respectively, indicating that the simulation performance of APM is effective.

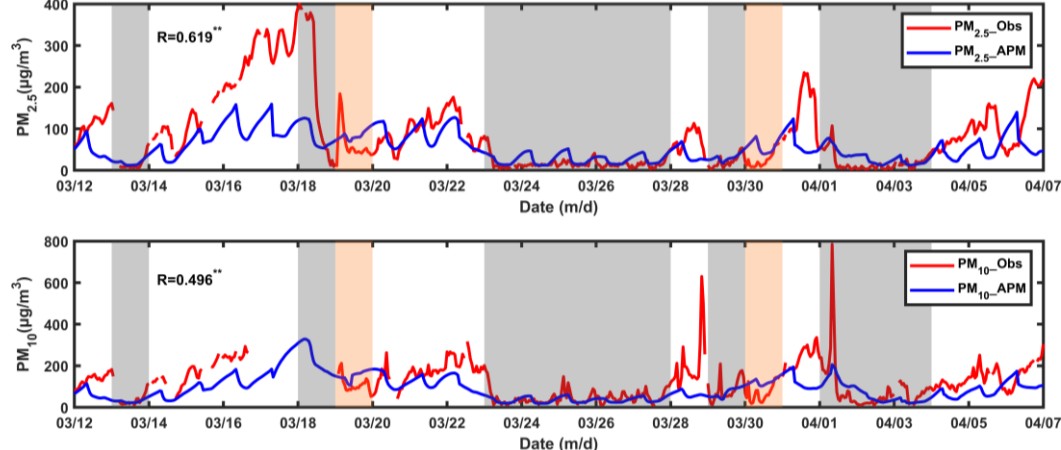

**Figure 13. Comparison of observation PM$_{2.5}$ and PM$_{10}$ mass concentrations with that in simulation based on APM model during the measurement period. The identified NPF days and undefined days are shadowed by grey and orange background, respectively.**

** the correlation coefficient passes the statistical significant test (p < 0.01).

In summary, in order to evaluate the ability of GEOS-Chem/APM and improve our understanding on nucleation

mechanisms, four nucleation schemes were chosen for the model. The BHN scheme and BIMN scheme underestimated the

observed particle number concentrations and failed to show significant number concentration fluctuation to distinguish NPF

event and non-event event days. The THN scheme well simulated the particle number concentrations on most NPF days except

March 18th and April 1st. The TIMN nucleation scheme can overall well simulate the total and sub-3 nm particle number

concentrations and nucleation rates in Beijing, which provides a basis for discussing the new particle nucleation mechanism

in Beijing. Besides, further understanding of the new particle nucleation mechanism in Beijing requires more meteorological

and precursor gas concentration data in other seasons and model evaluation based on other schemes. The effect of

meteorological conditions and precursors on nucleation can be further explored in sensitivity tests focusing on nucleation.

## 4 Summary and conclusions

During March 12th to April 6th, 2016 in Beijing, there were 11 typical NPF event days, 13 non-event days and 2 undefined

days. The sulfuric acid concentration in Beijing is sufficient to lead to NPF. Our study confirmed that low RH is indeed

favorable to the occurrence of an NPF event, as found by previous studies. From the perspective of organic compounds and

physical mechanism, we proposed some possible explanations for NPF events generally occur under the condition of strong

solar radiation. Low RH and $PM_{2.5}$ mass concentrations reduce condensation sink of condensable gases and coagulation

scavenging of new clusters. In contrast, high $PM_{2.5}$ concentrations and RH will increase the sinks of vapor responsible for

nucleation and growth of clusters.

Quantitative analysis indicates that when the daily total solar radiation was greater than 19 MJ/m$^2$/day and RH was less

than 26.5%, a new particle formation event would probably occur. However, the empirical condition from this case study was

possibly not applied to the general conditions, so it is necessary to conduct and examine more NPF events in Beijing and in

different seasons.

To understand the applicability of different nucleation schemes in polluted areas such as Beijing and improve our

understanding of new particle nucleation mechanisms, the simulations based on GEOS-Chem/APM model were conducted

during this observation period. It was found that the simulations with BHN and BIMN schemes systematically underestimated



both number concentrations and nucleation rates. TIMN scheme overall had a better performance than the THN scheme in terms of the simulations of the total and sub-3 nm particle number concentrations and nucleation rates. APM also reproduced the temporal variations of particle matter concentrations, indicating that the simulation performance of APM is effective.

It is acknowledged that more observations on NPF are required in order to further understand nucleation mechanisms in Beijing, especially long-term observational data, which is very important to examine the favorable conditions for nucleation events. More detailed and comprehensive comparisons between model predictions and relevant data obtained in various field measurements will help to further improve the understanding of nucleation mechanisms, explain observed nucleation events, and accurately predict air quality.

***Data availability.*** Data used in this work have been listed in Sect. 2.1 and acknowledgements.

***Author contributions.*** KW and XM developed the project idea. KW and RT designed and conducted the model experiments. KW analyzed the result and wrote the paper. XM and FY proposed scientific suggestions and revised the paper. All coauthors have read and commented on the manuscript.

***Competing interests.*** The authors declare that they have no conflict of interest.

***Acknowledgements.*** This study is supported by the National Key R&D Program of China grants (2019YFA0606802), the National Natural Science Foundation of China grants (42061134009 & 41975002). We are thankful to Prof. Jiang Jingkun for kindly providing new particle formation measurements. We are also grateful to GEOS-Chem Support Team for their management and maintenance of GEOS-Chem model.

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
