# Peer review of "Analysis of new particle formation events and comparisons to simulations of particle number concentrations based on GEOS-Chem/APM in Beijing, China"

_Atmospheric Chemistry and Physics, 2022_

## Author Comment (AC1)

**Response to the Comments of Referees**

**Journal:** Atmospheric Chemistry and Physics
**Manuscript Number:** acp-2022-797
**Title:** Analysis of new particle nucleation events and comparisons to simulations of particle number concentrations based on GEOS-Chem/APM in Beijing, China
**Author(s):** Kun Wang, Xiaoyan Ma[*], Rong Tian, Fangqun Yu

We thank the reviewers and editor for providing helpful comments to improve the manuscript. We have revised the manuscript according to the comments and suggestions of the referees.

The referee's comments are reproduced (black) along with our replies (blue). All the authors have read the revised manuscript and agreed with the submission in its revised form.

**< Anonymous Referee #1 >**

**Comment:** This study reanalyzed an NPF dataset reported in previous literature and explored possible nucleation mechanisms by contrasting measurements to simulations. It is interesting to see an attempt to reproduce urban NPF events with a global chemistry transport model. The topic of this study fits the scope of Atmospheric Chemistry and Physics. I recommend the authors take advantage of the review process to improve the manuscript substantially, such that it can meet the quality for publication.

We thank the referee for the positive comments on our manuscript. The manuscript has been carefully revised according to the referee's comments and suggestions.

**Major comments**

**Comment 1:** The TIMN scheme was found to be able to "overall well simulate the total and sub-3 nm particle number concentrations and nucleation rates in Beijing", with significantly higher values than BHN, BIHN, and THN. However, the nucleation rates in Fig. 10 seem to be lower than typical values in polluted megacities. For instance, the formation rates of freshly nucleated particles were usually higher than 10 cm-3 s-1 in Shanghai (Yao et al., 2018) and Beijing (Yan et al., 10.1029/2020GL091944).

The relatively coarse spatial resolution in a global model implies that the model produces a regional mean nucleation rate compared to the observation. Thus, it is difficult to perfectly reproduce the nucleation rate characteristics over urban areas. Nevertheless, we agree with the reviewer that nucleation rates measured in urban areas can be higher than $10 \ cm^{-3} \ s^{-1}$ (Yao et al., 2018; Yan et al., 2021) and this has been pointed out in the revised manuscript. The possible reasons for higher nucleation rates in urban areas are discussed in replies to the comments below.

Since the rate of ion-mediated nucleation may be limited by the ion production rate, as

well as the high sink, does this indicate that ion-mediated nucleation may not able to produce those high formation rates?

Yes. There are six parameters controlling nucleation rates for TIMN ($J_{TIMN}$): sulfuric acid vapor concentration ([$H_2SO_4$]), ammonia gas concentration ([$NH_3$]), temperature (T), relative humidity (RH), ionization rate (Q), and surface area of pre-existing particles (S) (Yu et al., 2020). Therefore, on the ground, the ion nucleation rates could be limited by ion production rates and cannot produce nucleation rates higher than Q. Compared to $J_{TIMN}$, there is one fewer controlling parameter for nucleation rates for THN ($J_{THN}$) (no Q dependence) so that $J_{THN}$ can be very high on the ground under certain conditions. Depending on the definition, THN may be treated as a part of TIMN in the ternary nucleation system (Yu et al., 2020). Nevertheless, we acknowledge that it is possible other nucleation mechanisms such as $H_2SO_4$-amine and $H_2SO_4$-organics nucleations may also simultaneously contribute to nucleation in polluted urban areas, which needs further study in the future.

Besides, it will be more convincing to show the measured particle formation rate in Fig. 10.

We have added the measured particle nucleation rates reported in Cai et al. (2017) to Figure R1.

[Figure]

**Figure R1. Time series of nucleation rates simulated on the basis of BHN, BIMN, THN, and TIMN schemes during the measurement period and the measured particle nucleation rates of 1.5 nm particles (Max $J_{1.5}$) reported in Cai et al. (2017). The identified NPF days and undefined days are shadowed by grey and orange background, respectively.**

These sentences have been added to the revised manuscript.

**Comment 2:** There might be a large room for improvement in the manuscript when addressing the current knowledge of NPF in terms of both nucleation mechanisms and their roles in the atmosphere. Some important advances in the last decade are missing from discussions.

Thanks for your suggestions. We have added more important findings to the introduction and discussion in the revised manuscript.

There are quite many places where the discussions are confusing and some are even at the risk of self-contradictory. Two examples are given below and some are given in minor comments, and I encourage the authors to improve the manuscript thoroughly.

Thanks for the suggestions and corrections, we have revised the manuscript accordingly (see our revised manuscript and response to minor comments).

Amines can be a key base for sulfuric acid nucleation in polluted megacities for their much higher efficiency in stabilizing clusters than ammonia and ions, as has been discussed in Yao et al. (2018) and many other studies. The authors have cited Yao et al. (2018) but why not address the roles of amines in the simulation?

We agree that amine can be a key base for sulfuric acid nucleation in polluted megacities. Recent measurements in urban Shanghai found that dimethylamine (DMA, $(CH_3)_2NH$) is perhaps the dominating base to stabilize $H_2SO_4$ clusters (Yao et al., 2018). Cai et al. (2021) found that $H_2SO_4$-amine nucleation can explain the observed high nucleation rate under the high coagulation sink. It's a good suggestion to address the roles of amines in the simulation. However, there is probably a long way to go before using the model to address the role of amines in nucleation for the following reasons. Firstly, the parameterization of sulfuric acid-amine nucleation scheme is not yet mature enough and a lot of validations against observations need to be done. Secondly, there is quite limited information on amine sources and thus all current emission inventories, to our knowledge, do not contain the inventories for amines. Therefore, it is not possible currently to carry out such simulations.
We have added more discussions to the revised manuscript.

Is it possible that sulfuric acid-amine nucleation can produce a comparable or higher nucleation rate than TIMN?

Yes, if concentrations of amines are high enough, sulfuric acid-amine nucleation can produce a comparable or higher nucleation rate than TIMN.

The authors stated that LVOCs play an important role in NPF, which is plausibly true and consistent with previous findings in Beijing. However, it is also stated that "TIMN scheme has a good simulation performance on the growth, condensation, coagulation and other processes after the nucleation process." Have the LVOCs been accounted for in TIMN? If not, does this indicates either a negligible contribution from LVOCs or a bias in the simulation results?

The TIMN does not include the contribution of nucleation involving LVOCs although

the contribution of LVOCs to growth is considered in the APM model. The growth of nucleated particles through the condensation of sulfuric acid vapor and equilibrium uptake of nitrate, ammonium, and secondary organic aerosol (SOA) is explicitly simulated, along with the scavenging of secondary particles by primary particles (dust, black carbon, organic carbon, and sea salt) (Yu and Luo, 2009). Yu (2011) has further developed the APM module to explicitly calculate the co-condensation of sulfuric acid and low-volatility secondary organic gases (LV-SOGs) or LVOCs on secondary particles and primary particles. The aerosol simulation considered the successive oxidation aging of the oxidation products of various VOCs (Yu, 2011). It should be noted that APM model contains the organics-mediated nucleation scheme (Yu et al., 2017) but it is not considered in the present study.

Sorry for the confusing statement. The difference in the growth process is due to different nucleation rates in the lookup tables of each scheme. It was inappropriate to state that "TIMN scheme has a good simulation performance on the growth, condensation, coagulation and other processes after the nucleation process." We have revised the model description chapter of the manuscript and explained the growth of nucleated particles in the model.

**Comment 3:** Most of the findings in the measurement part have been discussed in previous literature, which can also be seen from the discussions in the main text. The authors may need to clearly show the advances of this study compared to previous studies, including but not limited to the source of the dataset used in this study (Cai et al., 2017). Shortening the discussions, figures, and conclusions based on the measurement results can be an alternative way, and this will help emphasize the results based on simulations.

Thanks for the suggestions. For the measurements-based results, our revised manuscript mainly focused on the analysis of solar radiation and meteorological conditions favorable for NPF, and already shortened the discussions and deleted Figure 3.

**Minor comments**
**Comment 4:** Lines 55-56. This sentence is confusing because whether ions, specifically, charged particles measured by NAIS herein, grow faster than neutral particles or not is not directly relevant to the formation of the critical nucleus.

Thanks for your suggestions. This sentence has been deleted.

**Comment 5:** Lines 60-64. The mechanism proposed by Wu et al. (2020) is not a nucleation mechanism. It will be better to address it elsewhere, e.g., in lines 230-240.

Done as suggested.

**Comment 6:** Line 213, "7 % higher". It can be questionable to conclude the importance

of temperature on NPF based on a 7 % difference.

This sentence has been deleted.

**Comment 7:** Lines 296-297. Better to explain why and how a nucleation scheme can simulate the processes after nucleation.

The growth of nucleated particles through the condensation of sulfuric acid vapor and equilibrium uptake of nitrate, ammonium, and secondary organic aerosol (SOA) is explicitly simulated, along with the scavenging of secondary particles by primary particles (dust, black carbon, organic carbon, and sea salt) (Yu and Luo, 2009).
We have revised the model description chapter of the manuscript and explained the growth of nucleated particles in the model.

Sorry for the confusion. It was inappropriate to state that "TIMN scheme has a good simulation performance on the growth, condensation, coagulation and other processes after the nucleation process." This sentence has been deleted.

**Technical comments**
**Comment 8:** Lines 40-41. It is worth double-checking whether Huang et al. (2020) and Li et al. (2021) concluded that "new particles derived from NPF played a significant role in the formation of haze".

Corrected.

**Comment 9:** Line 130. Better to use steady-state or quasi-steady-state. NPF cannot reach an equilibrium.

Corrected.

**Comment 10:** Line 138, "banana shape". This is perhaps not necessary since NPF events in urban environments may not be banana-type events.

This sentence has been deleted.

**Comment 11:** Line 342. Please use "TIMN scheme" instead of "TIMN nucleation scheme", as N is for nucleation.

Corrected.

**Reference**

Cai, R., Yang, D., Fu, Y., Wang, X., Li, X., Ma, Y., Hao, J., Zheng, J., and Jiang, J.: Aerosol surface area concentration: a governing factor in new particle formation in

Beijing, Atmos. Chem. Phys., 17, 12327-12340, https://doi.org/10.5194/acp-17-12327-2017, 2017.

Cai, R., Yan, C., Yang, D., Yin, R., Lu, Y., Deng, C., Fu, Y., Ruan, J., Li, X., Kontkanen, J., Zhang, Q., Kangasluoma, J., Ma, Y., Hao, J., Worsnop, D. R., Bianchi, F., Paasonen, P., Kerminen, V.-M., Liu, Y., Wang, L., Zheng, J., Kulmala, M., and Jiang, J.: Sulfuric acid–amine nucleation in urban Beijing, Atmos. Chem. Phys., 21, 2457–2468, https://doi.org/10.5194/acp-21-2457-2021, 2021.

Yan, C., Yin, R., Lu, Y., Dada, L., Yang, D., Fu, Y., Kontkanen, J., Deng, C., Garmash, O., Ruan, J., Baalbaki, R., Schervish, M., Cai, R., Bloss, M., Chan, T., Chen, T., Chen, Q., Chen, X., Chen, Y., Chu, B., Dällenbach, K., Foreback, B., He, X., Heikkinen, L., Jokinen, T., Junninen, H., Kangasluoma, J., Kokkonen, T., Kurppa, M., Lehtipalo, K., Li, H., Li, H., Li, X., Liu, Y., Ma, Q., Paasonen, P., Rantala, P., Pileci, R. E., Rusanen, A., Sarnela, N., Simonen, P., Wang, S., Wang, W., Wang, Y., Xue, M., Yang, G., Yao, L., Zhou, Y., Kujansuu, J., Petäjä, T., Nie, W., Ma, Y., Ge, M., He, H., Donahue, N. M., Worsnop, D. R., Veli-Matti, K., Wang, L., Liu, Y., Zheng, J., Kulmala, M., Jiang, J., and Bianchi, F.: The Synergistic Role of Sulfuric Acid, Bases, and Oxidized Organics Governing New-Particle Formation in Beijing, Geophys. Res. Lett., 48, e2020GL091944, https://doi.org/10.1029/2020gl091944, 2021.

Yao, L., Garmash, O., Bianchi, F., Zheng, J., Yan, C., Kontkanen, J., Junninen, H., Mazon, S., Ehn, M., Paasonen, P., Sipilä, M., Wang, M., Wang, X., Xiao, S., Chen, H., Lu, Y., Zhang, B., Wang, D., Fu, Q., Geng, F., Li, L., Wang, H., Qiao, L., Yang, X., Chen, J., Kerminen, V., Petäjä, T., Worsnop, D., Kulmala, M., and Wang, L.: Atmospheric new particle formation from sulfuric acid and amines in a Chinese megacity, Science, 361, 278-281, https://doi.org/10.1126/science.aao4839, 2018.

Yu, F.: A secondary organic aerosol formation model considering successive oxidation aging and kinetic condensation of organic compounds: global scale implications, Atmos. Chem. Phys., 11, 1083–1099, https://doi.org/10.5194/acp-11-1083-2011, 2011.

Yu, F. and Luo, G.: Simulation of particle size distribution with a global aerosol model: contribution of nucleation to aerosol and CCN number concentrations, Atmos. Chem. Phys., 9, 7691-7710, https://doi.org/10.5194/acp-9-7691-2009, 2009.

Yu, F., Luo, G., Nadykto, A. B., and Herb, J.: Impact of temperature dependence on the possible contribution of organics to new particle formation in the atmosphere, Atmos. Chem. Phys., 17, 4997–5005, https://doi.org/10.5194/acp-17-4997-2017, 2017.

Yu, F., Nadykto, A. B., Luo, G., and Herb, J.: $H_2SO_4$–$H_2O$ binary and $H_2SO_4$–$H_2O$–$NH_3$ ternary homogeneous and ion-mediated nucleation: lookup tables version 1.0 for 3-D modeling application, Geosci. Model Dev., 13, 2663–2670, https://doi.org/10.5194/gmd-13-2663-2020, 2020.

---

## Author Comment (AC2)

**Response to the Comments of Referees**

**Journal:** Atmospheric Chemistry and Physics
**Manuscript Number:** acp-2022-797
**Title:** Analysis of new particle nucleation events and comparisons to simulations of particle number concentrations based on GEOS-Chem/APM in Beijing, China

**Author(s):** Kun Wang, Xiaoyan Ma[*], Rong Tian, Fangqun Yu

We thank the reviewers and editor for providing helpful comments to improve the manuscript. We have revised the manuscript according to the comments and suggestions of the referees.

The referee's comments are reproduced (black) along with our replies (blue). All the authors have read the revised manuscript and agreed with the submission in its revised form.

**< Anonymous Referee #2 >**

**Comment:** This study simulates NPF events in Beijing by applying GEOS-Chem/APM model, considering four nucleation mechanisms. It improved our understanding on nucleation and influence by meteorological factors. The TIMN nucleation scheme can predict nucleation well, however, more direct measurement data needed for validation the modeling results. The effect of meteorological conditions and precursors on nucleation should be further discussed in details. This paper is well organized and written. I recommend it can be accepted after the following major revisions.

We thank the referee for the positive comments on our manuscript. The manuscript has been carefully revised according to the referee's comments and suggestions.

**Major issues**

**Comment 1:** Line 113, The nucleation mechanism used in APM model is IMN parameterization scheme, which is developed based on the measurement and laboratory data elsewhere (Yu, 2006b). However, whether the parameterization is applicable in urban Beijing, with the high pollution level and unclear role that organics take place?

The reviewer raised a good point. Nucleation is a key process controlling particle properties in the atmosphere. To better understand the formation and evolution mechanisms of air pollution, especially in heavy pollution areas, it is necessary to assess the applicability of nucleation parametrizations currently available. This study aims to evaluate the performance of four currently widely-used nucleation schemes, and provide some clues on the contribution of different nucleation pathways to aerosol number concentrations.

The nucleation rates of different schemes in this study were calculated using lookup tables, which captured well the absolute values of nucleation rates and their dependence on key controlling parameters as observed during the well-controlled Cosmics Leaving

Outdoor Droplets (CLOUD) experiments (Yu et al., 2020). According to our simulations, the parameterization schemes can capture some new particle formation events in urban Beijing and provide a basis for discussing the new particle nucleation mechanism in urban areas. We acknowledge that it is possible other nucleation mechanisms such as $H_2SO_4$-amine and $H_2SO_4$-organics nucleations may also simultaneously contribute to nucleation in polluted urban areas, which needs further study in the future.

Can you talk about the uncertainties or bias of the simulation result due to the four parameterizations?

Some uncertainties may exist in nucleation schemes as a result of uncertainties in the thermodynamics data used in the nucleation model. There are six parameters controlling nucleation rates for TIMN ($J_{TIMN}$), including sulfuric acid vapor concentration ($[H_2SO_4]$), ammonia gas concentration ($[NH_3]$), temperature (T), relative humidity (RH), ionization rate (Q), and surface area of pre-existing particles (S). Compared to $J_{TIMN}$, there is one fewer controlling parameter for nucleation rates for THN ($J_{THN}$) (no Q dependence) and BIMN ($J_{BIMN}$) (no $[NH_3]$ dependence), while nucleation rate for BHN ($J_{BHN}$) only depends on four parameters ($[H_2SO_4]$, T, RH, and S). The uncertainties in the values of these parameters simulated by the model, as a result of uncertainties in the emissions, chemistry, and meteorology, will affect the simulation results. In addition, the real nucleation mechanisms in the atmosphere are complex and may involve additional parameters. Besides, the comparison between simulation results based on large grids and measurements also created uncertainties. These sentences have been added to the revised manuscript.

**Comment 2:** Figure 2, can you explain why there no clear difference of sulfuric acid concentration between NPF and non-NPF days. Table 1 is not necessary as only two numbers are given. It can be given in the text, and better to give the mean ± standard deviation.

Thanks for the suggestions, we have deleted Table 1 and corrected the mean ± standard deviation.
According to Cai et al. (2017), sulfuric acid in Beijing during the campaign period was sufficiently high for nucleation events to occur and NPF events appeared to be governed by aerosol Fuchs surface area. The presence of gaseous sulfuric acid in concentrations exceeding $10^5$ molecules $cm^{-3}$ has been shown as a necessary condition to occur NPF in the atmosphere (Weber et al., 1999; Nieminen et al., 2009). Yan et al. (2021) found that observed sulfuric acid concentrations were higher on non-NPF days than on NPF days in the winter of 2018. Therefore, in this study, it is reasonable that sulfuric acid concentrations on NPF and non-NPF days were close.

In addition, as the authors mentioned, the sulfuric acid reported in this study is lower than the other studies, can you give the concentration level given by other studies? As

SO$_2$ decreased recent year in Beijing, the comparison should be conducted at the recent years, and also differed by seasons.

The daily maximum sulfuric acid concentrations measured in this study ($> 10^6$ cm$^{-3}$) are lower than those in summer of 2017 in Beijing ($> 10^7$ cm$^{-3}$) (Wu et al., 2020) and close to those in winter of 2019 in Beijing ($> 10^6$ cm$^{-3}$) (Foreback et al., 2022). This might be caused by the relatively weak solar radiation intensity in springtime and wintertime measurements compared with summertime observation.
We have revised the corresponding content of the manuscript.

**Comment 3:** Line 214, some studies reported that temperature can influence the NH$_3$ stabilizing with sulfuric acid, which finally affect the nucleation rate. However, in this work, it cannot be concluded the roles of temperature. In Beijing, NPF occurs more frequent in spring, winter than summer. The higher temperature on NPF days probably related with stronger solar radiation on clear days. It is difficult to evaluate the roles of temperature, as temperature, RH and solar radiation correlated under the similar synoptic conditions. As well as in line 228, I don't think a simple metrological factor, RH or solar radiation, can explain the NPF reasonably (such as, high RH usually occurs under cloudy days with low solar radiation). The meteorological factors have systematically influence on NPF.

Thanks for the comments. We agree that meteorological factors have systematical influences on NPF and it is difficult to isolate the effect of single factor on NPF.
We have revised the corresponding content of the manuscript and deleted Figures 4 and 5.

**Comment 4:** Figure 7, can you explain why modeled RH is much lower than the observed value? It this reasonable with the model uncertainties? For example, on March 26 and 27, the bias can be 10 K.

Temperatures (at the first layer, about 70 meters above the surface) input to the model are significantly lower than measurements taken near the surface. We compared temperatures (at 10 m above the displacement height) input to the model with measurements. As shown in Figure R1, the temporal variations of the temperature input to the model were more consistent with observations than before. Besides, the coarse spatial resolution may also cause the bias because of the "urban heat island" effect.

We have revised the Figure as Figure R1.

[Figure]

**Figure R1. Time series of temperature (at 10 m above the displacement height) and RH (at the first layer, about 70 meters) input to the model and time series of measurements at a height of about 10 meters during the 26-day campaign.**

**Comment 5:** Figure 10, can you calculate the observed nucleation rate, as compared with the simulated nucleation rate.

Thanks for your suggestion. We have added the measured particle nucleation rates reported in Cai et al. (2017) to Figure R2. The simulated nucleation rates in Figure R2 were lower than measured particle nucleation rates reported in Cai et al. (2017). The relatively coarse spatial resolution in a global model implies that the model produces a regional mean nucleation rate compared to the observation. Thus, it is difficult to perfectly reproduce the nucleation rate characteristics over urban areas. Moreover, we acknowledge that it is possible other nucleation mechanisms such as $H_2SO_4$-amine and $H_2SO_4$-organics nucleations may also simultaneously contribute to nucleation in polluted urban areas, which needs further study in the future.

[Figure]

**Figure R2. Time series of nucleation rates simulated on the basis of BHN, BIMN, THN, and TIMN schemes during the measurement period and the measured particle nucleation rates of 1.5 nm**

**particles (Max J$_{1.5}$) reported in Cai et al. (2017). The identified NPF days and undefined days are shadowed by grey and orange background, respectively.**

Figure 11, the vertical distribution of nucleation rate has large uncertainties, and even no vertical data can validate the model result. I don't think it is robust confidence to represent the nucleation rates in the upper boundary layer in line 297.

We have deleted Figure 11 and the related discussion.

**Comment 6:** Can you model the sulfuric acid and validate the results by the measurement data? This can improve the confidence of model results.

Thanks for your suggestion. We believe that it may be more reasonable to compare the simulated sulfuric acid concentrations by higher resolution with the field observations. Because the simulated sulfuric acid concentration in a large grid cannot represent the measured sulfuric acid at a site in urban Beijing. We can discuss this problem in future work.

**Minor problems:**
**Comment 7:** Line 31-34, First, "new particle nucleation" is a repetitive phrase, normally we call this phenomenon "new particle formation", which includes nucleation and growth process. For the second sentence, the nucleated particles undergone condensation and coagulation processes and grow into larger sizes. However, water absorption is an independent process that characterize the particle hygroscopicity, which should not be included in the new particle formation process.

Thanks. Revised as suggested.

**Comment 8:** Line 99 and 101, for the data sources from website, the latest access time should be given.

Corrected.

**Comment 9:** Figure 1, the contour plot of PNSD near the detection limit below 5 nm looks wired, it seems only data of NPF was given. How is the data on other days? It is zero or has been excluded from the dataset? The author should provide the details about how to handle the PNSD data.

Sorry for the confusion. The plot near the detection limit below 3 nm in Figure 1 looks wired, possibly because the instrument for measuring sub-3 nm particles is different from the instrument for observing 3 nm to 10 μm (which is beyond the scope of this study).
We did not exclude the data on other days. Except for NPF days, the number concentrations of sub-3 nm particles on other days are very low and almost zero.

**Comment 10:** Line 213, 8 to 16:00 UTC or Local time, o'clock is not a formal written language.

Corrected.

**Comment 11:** Table 2, the restricted conditions (RH and solar radiation) proposed for identifying NPF only based on one-month measurement data is not robust. Even at the same location, the criterion can be changed due to seasonal variation of meteorological factors.

Thanks for the comment which we agree. We hope to conduct such analysis once more measurement data are available in the future, in order to improve our understanding. Table 2 has been deleted.

**Comment 12:** Figure 13, APM model cannot capture the peaks of $PM_5$ and $PM_{10}$, especially the severe pollution episode from March 14 to 18, is this due to the model spatial resolution or the emission inventory uncertainty?

Yes, we think that both the model spatial resolution and the emission inventory are potential factors to affect the results. We will discuss these issues in the future study. We have added these discussions to the revised manuscript.

**Reference**

Cai, R., Yang, D., Fu, Y., Wang, X., Li, X., Ma, Y., Hao, J., Zheng, J., and Jiang, J.: Aerosol surface area concentration: a governing factor in new particle formation in Beijing, Atmos. Chem. Phys., 17, 12327-12340, https://doi.org/10.5194/acp-17-12327-2017, 2017.

Foreback, B., Dada, L., Daellenbach, K. R., Yan, C., Wang, L., Chu, B., Zhou, Y., Kokkonen, T. V., Kurppa, M., Pileci, R. E., Wang, Y., Chan, T., Kangasluoma, J., Zhuohui, L., Guo, Y., Li, C., Baalbaki, R., Kujansuu, J., Fan, X., Feng, Z., Rantala, P., Gani, S., Bianchi, F., Kerminen, V.-M., Petäjä, T., Kulmala, M., Liu, Y., and Paasonen, P.: Measurement report: A multi-year study on the impacts of Chinese New Year celebrations on air quality in Beijing, China, Atmos. Chem. Phys., 22, 11089–11104, https://doi.org/10.5194/acp-22-11089-2022, 2022.

Nieminen, T., Manninen, H. E., Sihto, S. L., Yli-Juuti, T., Mauldin, I. R. L., Petaja, T., Riipinen, I., Kerminen, V. M., and Kulmala, M.: Connection of sulfuric acid to atmospheric nucleation in boreal forest, Environ. Sci. Technol., 43(13), 4715-4721, https://doi.org/10.1021/es803152j, 2009.

Weber, R. J., McMurry, P. H., Mauldin, R. L., Tanner, D. J., Eisele, F. L., Clarke, A. D.,

and Kapustin, V. N.: New particle formation in the remote troposphere: a comparison of observations at various sites, Geophys. Res. Lett., 26(3), 307-310, https://doi.org/10.1029/1998gl900308, 1999.

Wu, H., Li, Z., Li, H., Luo, K., Wang, Y., Yan, P., Hu, F., Zhang, F., Sun, Y., Shang, D., Liang, C., Zhang, D., Wei, J., Wu, T., Jin, X., Fan, X., Cribb, M., Fischer, M., Kulmala, M., and Petäjä, T.: The impact of the atmospheric turbulence-development tendency on new particle formation: a common finding on three continents, National Science Review, 8(3), 140-150, https://doi.org/10.1093/nsr/nwaa157, 2020.

Yan, C., Yin, R., Lu, Y., Dada, L., Yang, D., Fu, Y., Kontkanen, J., Deng, C., Garmash, O., Ruan, J., Baalbaki, R., Schervish, M., Cai, R., Bloss, M., Chan, T., Chen, T., Chen, Q., Chen, X., Chen, Y., Chu, B., Dällenbach, K., Foreback, B., He, X., Heikkinen, L., Jokinen, T., Junninen, H., Kangasluoma, J., Kokkonen, T., Kurppa, M., Lehtipalo, K., Li, H., Li, H., Li, X., Liu, Y., Ma, Q., Paasonen, P., Rantala, P., Pileci, R. E., Rusanen, A., Sarnela, N., Simonen, P., Wang, S., Wang, W., Wang, Y., Xue, M., Yang, G., Yao, L., Zhou, Y., Kujansuu, J., Petäjä, T., Nie, W., Ma, Y., Ge, M., He, H., Donahue, N. M., Worsnop, D. R., Veli-Matti, K., Wang, L., Liu, Y., Zheng, J., Kulmala, M., Jiang, J., and Bianchi, F.: The Synergistic Role of Sulfuric Acid, Bases, and Oxidized Organics Governing New-Particle Formation in Beijing, Geophys. Res. Lett., 48, e2020GL091944, https://doi.org/10.1029/2020gl091944, 2021.

Yu, F., Nadykto, A. B., Luo, G., and Herb, J.: $H_2SO_4$–$H_2O$ binary and $H_2SO_4$–$H_2O$–$NH_3$ ternary homogeneous and ion-mediated nucleation: lookup tables version 1.0 for 3-D modeling application, Geosci. Model Dev., 13, 2663–2670, https://doi.org/10.5194/gmd-13-2663-2020, 2020.

---

## Referee Report (RR1)

The manuscript has improved a lot and been corrected according to the comments. All the issues raised by the reviewers has been addressed point by point. I recommend this paper can be accepted after the minor revisions as given below:

1. The authors have fully discussed the limitation and uncertainties of the model result in this study. as you mentioned the H2SO4-DMA nucleation is probably the major path way of nucleation in Beijing (Yan et al., 2021; Cai et al.,2021,2022), and in your work, TIMN scheme simulation agree well with the nucleation. So how is the atmospheric implication of TIMN in Beijing? And have evaluated the contribution by "ion-mediated process" in Beijing?

2. Line 17, to be favorable for

3. Line 55-56, please add the references to support the first sentence in this paragraph.

4. Line 60-63, can you give the suitable conditions for each nucleation mechanism, as you mentioned $HIO_3$ nucleation is dominant for coastal areas.

5. Line 188-189, I don't think the resolution of sulfuric acid can be 1 molecule/cm³, I recommend the digitals are given as $(6.1\pm3.1)*10^5$

6. Line 207, governed by aerosol Fuchs surface area, which is a representative parameter of coagulation scavenging (Cai et al., 2017a).

7. Line 216, necessary condition for NPF occurrence in the atmosphere

8. Line 329, can not

9. Line 420-425, does the APM model consider the nucleated particle growth process when simulate the PM mass concentration? If it is possible to evaluated the contribution of NPF to particle mass?

---

## Author Response (AR2)

**Response to the Comments of Referee**

**Journal:** Atmospheric Chemistry and Physics

**Manuscript Number:** acp-2022-797

**Title:** Analysis of new particle formation events and comparisons to simulations of particle number concentrations based on GEOS-Chem/APM in Beijing, China

**Author(s):** Kun Wang, Xiaoyan Ma[*], Rong Tian, Fangqun Yu

We thank the reviewer and editor for providing helpful comments to improve the manuscript. We have revised the manuscript according to the comments and suggestions of the referee.

The referee's comments are reproduced (black) along with our replies (blue). All the authors have read the revised manuscript and agreed with the submission in its revised form.

**< Anonymous Referee >**

**Comment:** The manuscript has improved a lot and been corrected according to the comments. All the issues raised by the reviewers has been addressed point by point. I recommend this paper can be accepted after the minor revisions as given below.

Thanks for the suggestions and corrections, we have revised the manuscript accordingly.

**Comment 1:** The authors have fully discussed the limitation and uncertainties of the model result in this study. as you mentioned the $H_2SO_4$-DMA nucleation is probably the major path way of nucleation in Beijing (Yan et al., 2021; Cai et al.,2021,2022), and in your work, TIMN scheme simulation agree well with the nucleation. So how is the atmospheric implication of TIMN in Beijing?

In real atmosphere, there is probably more than one nucleation mechanisms occurring in some cases. Overall, TIMN agree better than other nucleation schemes used in this study, but one can easily see that TIMN simulation obviously underestimated the observed nucleation rate on March 13th (> 150 $cm^{-3}$ $s^{-1}$), which may imply that other nucleation schemes which have not been included in the simulations, e.g., $H_2SO_4$-DMA or $H_2SO_4$-organics, may also simultaneously contribute to nucleation in this case.

And have evaluated the contribution by "ion-mediated process" in Beijing?

Sorry for misleading. We meant that four nucleation schemes including ion-mediated nucleation have been evaluated.

**Comment 2:** Line 17, to be favorable for

Corrected.

**Comment 3:** Line 55-56, please add the references to support the first sentence in this paragraph

Thanks for the suggestions. We have added references in the revised manuscript.

**Comment 4:** Line 60-63, can you give the suitable conditions for each nucleation mechanism, as you mentioned $HIO_3$ nucleation is dominant for coastal areas.

Thanks for the suggestions. $H_2SO_4$-$H_2O$ binary theory usually predicts the nucleation rates at low temperatures, high relative humidities, small pre-existing aerosol concentrations and high sulfuric acid concentrations (Kulmala et al., 1998, 2000). For $NH_3$ mixing ratios exceeding about 1 ppt, the $H_2SO_4$-$NH_3$-$H_2O$ ternary nucleation enhances the binary $H_2SO_4$-$H_2O$ nucleation rate by several orders of magnitude (Korhonen et al., 1999). Ion-mediated nucleation mechanism provides a consistent explanation for a variety of tropospheric observations (Yu and Turco, 2000). Organics-mediated nucleation can explain NPF in some polluted areas (Wang et al., 2015).
These sentences have been added to the revised manuscript.

**Comment 5:** Line 188-189, I don't think the resolution of sulfuric acid can be 1 molecule/cm3, I recommend the digitals are given as $(6.1\pm3.1)*10^5$

Done as suggested.

**Comment 6:** Line 207, governed by aerosol Fuchs surface area, which is a representative parameter of coagulation scavenging (Cai et al., 2017a).

Corrected.

**Comment 7:** Line 216, necessary condition for NPF occurrence in the atmosphere

Corrected.

**Comment 8:** Line 329, can not

Corrected.

**Comment 9:** Line 420-425, does the APM model consider the nucleated particle growth process when simulate the PM mass concentration? If it is possible to evaluated the contribution of NPF to particle mass?

The APM model consider the contribution of nucleated particle growth process to total PM mass concentration.

In the atmosphere, the surface area of pre-existing particles not only serves as a coagulation sink but also as a condensation sink for precursor gases. The APM model takes into account the effect of surface area of pre-existing particles (Yu et al., 2020). In the model, the precursor gases that do not involve in the growth of nucleated particles (for cases with very small nucleation rates) condense on pre-existing particles instead so $PM_{2.5}$ and $PM_{10}$ mass concentrations are close for all four schemes.

These sentences have been added to the revised manuscript.

**Reference**

Korhonen, P., Kulmala, M., Laaksonen, A., Viisanen, Y., McGraw, R., and Seinfeld, J.: Ternary nucleation of $H_2SO_4$, $NH_3$, and $H_2O$ in the atmosphere, J. Geophys. Res., 104, 26349-26353, https://doi.org/10.1029/1999jd900784, 1999.

Kulmala, M., Laaksonen, A., and Pirjola, L.: Parameterizations for sulfuric acid/water nucleation rates, J. Geophys. Res., 103, 8301-8307, https://doi.org/10.1029/97jd03718, 1998.

Kulmala, M., Pirjola, L., and Mäkelä, J.: Stable sulphate clusters as a source of new atmospheric particles, *Nature*, 404, 66–69, https://doi.org/10.1038/35003550, 2000.

Wang, Z., Hu, M., Pei, X., Zhang, R., Paasonen, P., Zheng, J., Yue, D., Wu, Z., Boy, M., and Wiedensohler, A.: Connection of organics to atmospheric new particle formation and growth at an urban site of Beijing, Atmos. Environ., 103, 7-17, https://doi.org/10.1016/j.atmosenv.2014.11.069, 2015.

Yu, F. and Turco, R.: Ultrafine aerosol formation via ion-mediated nucleation, Geophys. Res. Lett., 27(6), 883-886, https://doi.org/10.1029/1999gl011151, 2000.

Yu, F., Nadykto, A. B., Luo, G., and Herb, J.: $H_2SO_4$–$H_2O$ binary and $H_2SO_4$–$H_2O$–$NH_3$ ternary homogeneous and ion-mediated nucleation: lookup tables version 1.0 for 3-D modeling application, Geosci. Model Dev., 13, 2663–2670, https://doi.org/10.5194/gmd-13-2663-2020, 2020.